# Electrical detection and modulation of magnetism in a Dy-based ferroelectric single-molecule magnet

Yu-Xia Wang [1,4], Dan Su[2,4], Yinina Ma[2], Young Sun [2,3] ✉ & Peng Cheng [1] ✉

Electrical control of magnetism in single-molecule magnets with peculiar quantum magnetic behaviours has promise for applications in molecular electronics and quantum computing. Nevertheless, such kind of magneto-electric effects have not been achieved in such materials. Herein, we report the successful realization of significant magnetoelectric effects by introducing ferroelectricity into a dysprosium-based single-molecule magnet through spatial cooperation between flexible organic ligands and halide ions. The stair-shaped magnetization hysteresis loop, alternating current susceptibility, and magnetic relaxation can be directly modulated by applying a moderate electric field. Conversely, the electric polarization can be modulated by applying a small magnetic field. In addition, a resonant magnetodielectric effect is clearly observed, which enables detection of quantum tunnelling of magnetization by a simple electrical measurement. The integration of ferroelectricity into single-molecule magnets not only broadens the family of single-molecule magnets but also makes electrical detection and modulation of the quantum tunnelling of magnetization a reality.

Single-molecule magnets (SMMs) constitute a special class of metal-organic hybride magnets that are characterized by magnetic bistability with resonant quantum tunneling of magnetization (RQTM)[1,2]. For more than two decades, considerable effort has been devoted to the design and synthesis of SMMs with higher magnetic blocking temperatures and greater coercivity to fit potential applications in molecular spintronics and quantum computing[3-9]. In contrast to the well-studied magnetic properties, the electrical properties of SMMs have been ignored because these materials are usually good insulators. As a result, conventional electrical transport properties such as the magnetoresistance and the Hall effect are not measurable in insulating SMMs. Nevertheless, one may anticipate that ferroelectricity can be introduced into insulating SMMs to endow them with more appealing dielectric and ME

properties. This strategy would yield ferroelectric SMMs in which single-molecule quantum magnetism coexists with ferroelectricity.

The ME effects have drawn considerable interest over the past two decades, with a focus on multiferroic materials where two or more ferroic orders coexist[10-13]. A large number of multiferroic materials have been discovered and the majority of them are transition metal oxides[14-18]. There has been increasing effort in the search for new families of magnetoelectric multiferroics beyond conventional inorganic oxides. For example, metal–organic frameworks consisting of magnetic metal ions and polar organic groups can become multiferroic and exhibit clear ME effects[19-22]. Furthermore, organic molecular ferroelectrics containing magnetic ions may provide an alternative way to produce the ME effects[23-25].

[1]Key Laboratory of Advanced Energy Material Chemistry, Frontiers Science Center for New Organic Matter, Renewable Energy Conversion and Storage Center, and Haihe Laboratory of Sustainable Chemical Transformations (Tianjin), College of Chemistry, Nankai University, Tianjin 300071, China. [2]Beijing National Laboratory for Condensed Matter Physics, Institute of Physics, Chinese Academy of Sciences, 100190 Beijing, China. [3]Department of Applied Physics and Center of Quantum Materials and Devices, Chongqing University, Chongqing 401331, China. [4]These authors contributed equally: Yu-Xia Wang, Dan Su. ✉e-mail: youngsun@cqu.edu.cn; pcheng@nankai.edu.cn

In a previous study, a weak magnetodielectric (MD) effect was observed in a nonferroelectric dysprosium-based SMM[26]. If ferroelectricity can be introduced into SMMs, the coupling between magnetism and electricity could be greatly enhanced. Therefore, one may expect pronounced ME effects in ferroelectric SMMs, which will pave a way for device applications using SMMs.

Herein, to achieve the ME effects in SMMs, we designed and synthesized a dysprosium-based SMM, $[Dy(L)_2(H_2O)_5]Cl_3 \cdot H_2O \cdot CH_3CN$ (Dy-SMM, L = $^tBuPO(NH^iPr)_2$) with boundary hydrogen bonds between the flexible organic ligands and the halide ions. This SMM crystallizes in the nonpolar space group $P2_1/c$ at room temperature but transforms into the polar space group $P2_1$ at low temperatures. Ferroelectricity as well as strong ME effects are demonstrated in millimeter-sized single crystals. A low magnetic field is able to suppress electric polarization, and the magnetic properties are effectively modulated by applying electric fields. Moreover, a resonant MD effect is observed, which enables electrical detection of RQTM in the SMM.

## Results

### Structure of the Dy-SMM

Millimeter-sized single crystals of Dy-SMM were successfully prepared to satisfy the requirements for structural characterization and magnetic/electrical measurements. The results of single-crystal X-ray diffraction at different temperatures revealed that Dy-SMM crystallized into the monoclinic asymmetric polar space group $P2_1$ at 50 K but into the monoclinic symmetric space group $P2_1/c$ at 300 K (Fig. 1a and Supplementary Table 1). The coordination environment of the $Dy^{3+}$ ion is a slightly distorted pentagonal bipyramid geometry. Continuous shape measure (CSM) calculations were carried out on the $DyO_7$ sites (Supplementary Table 2)[27]. Detailed crystal description is provided in supplementary discussion (Supplementary Figs. 1–3).

Luminescence spectrum of Dy-SMM in the temperature range of 5–295 K was studied (Supplementary Fig. 4). The luminescence intensity decreases with increasing temperature, but shows a sudden rise above ~200 K. At low temperatures, the peak at $\lambda = 568$ nm was dominant. In contrast, the peak at $\lambda = 578$ nm becomes dominant at high temperatures. These changes in luminescence spectra provide further evidence for a structural phase transition in Dy-SMM with increasing temperature.

As breaking of inversion symmetry is a prerequisite for introducing ferroelectricity, we analyzed the correlation between the temperature-induced symmetry transition and structural changes. The structure at 300 K suggests that the $c$ glide planes are across the $^tBu$ groups from one of the two coordinated ligands. By comparing the nearby diagonal $^tBu$ groups, we found that the P–C and C–C bond lengths in the ligands increases or decreases slightly at 50 K, and hence, the structure is no longer symmetric. The nonpolar-to-polar structural transition indicates that Dy-SMM is ferroelectric at low temperatures.

### Ferroelectricity characterization

To study the ferroelectricity of Dy-SMM, the dielectric and pyroelectric properties were investigated within the temperature range of 2–300 K. The dielectric permittivity ($\varepsilon_r$) and loss tangent (tan δ) perpendicular to the (0–11) plane exhibited a pronounced peak at ~250 K, indicating a paraelectric-to-ferroelectric phase transition (Fig. 1b). A pyroelectric current peak was also observed at ~250 K, consistent with the dielectric anomaly (Supplementary Fig. 5), providing direct evidence of ferroelectricity. We note that there is no extra dielectric peak between 2 and 250 K, which suggests that no other structural transition occurs at low temperatures. Therefore, the ferroelectricity persists down to the lowest temperature studied.

The electric polarization ($P$) was determined by integrating the pyroelectric current over time after subtracting the background. The electric polarization under positive and negative electric fields is reversible (Fig. 1c), and the value of polarization (~2 μC/cm²) is comparable to those of conventional ferroelectrics[28]. Hence, Dy-SMM was confirmed to exhibit ferroelectricity below ~250 K.

### Single-molecule magnet characterization

To investigate the SMM properties of Dy-SMM, the AC magnetic susceptibilities of both powder and single-crystal samples were measured. Detailed magnetic relaxation description is presented in supplementary discussion. Both the in-phase ($\chi'$) and out-of-phase ($\chi''$) components showed a strong dependence on temperature (Supplementary Fig. 6) and frequency (Fig. 2a and Supplementary Fig. 7), in accordance with the intrinsic magnetization dynamics for SMMs. To analyze the magnetic relaxation processes of the single-crystal sample of Dy-SMM, the generalized Debye model was used to fit the Cole–Cole plots[29] of the AC magnetic susceptibility (Supplementary Fig. 8). The temperature dependence of relaxation time yielded $U_{eff} = 643(4)$ and $601(1)$ K for $H$ perpendicular and parallel to the (0–11) plane, respectively (a detailed magnetic relaxation discussion is presented in supplementary discussion). In addition, the results agree well with the ab initio calculations which yielded an energy barrier of 677 K (Supplementary Table 6), and illustrate that the Orbach process likely proceeded via the second excited state (Supplementary Figs. 9 and 10).

### Magnetic anisotropy

Owing to an ideal model of the $D_{5h}$ geometry, Dy-SMM is expected to have high-performance SMM properties due to the appropriate uniaxial anisotropy. The zero-field cooling (ZFC) and field cooling (FC) magnetizations were measured along two directions: $H$ perpendicular and parallel to the (0–11) plane (Supplementary Fig. 11a). For $H$ parallel to the (0–11) plane, there is no clear divergence between the ZFC and FC magnetization down to 2 K. However, for $H$ perpendicular to the (0–11) plane, divergence between the ZFC and FC magnetization occurs below the magnetic blocking temperature of ~12 K. These results demonstrate the strong magnetic anisotropy and the characteristic magnetic blocking behavior of Dy-SMM.

Regular stair-shaped magnetic hysteresis loops at low temperatures, characteristic of RQTM, were obtained in Dy-SMM (Fig. 2b). For $H$ perpendicular to the (0–11) plane, the hysteresis loops are wide, with a coercive field $H_C = 14$ kOe (2 K, 30 Oe/s). In contrast, the coercivity is nearly zero when $H$ is parallel to the (0–11) plane. The shape of the hysteresis loop also depends on the sweep rate of the magnetic field. The magnetization step decreases and the hysteresis loop broadens with increasing sweep rate (Supplementary Fig. 11b). To determine the easy axis of the magnetic anisotropy, we measured the magnetization with sample rotation (Supplementary Fig. 12). The results demonstrate a uniaxial anisotropy, agreeing with $D_{5h}$ geometry of the Dy-SMM. The easy axis is 87.5 degree perpendicular to the (0–11) plane, i.e., close to the [0–11] direction. Therefore, due to the coexistence of ferroelectricity and SMM behaviors, Dy-SMM represents a prototype of ferroelectric SMMs.

### Magnetodielectric and direct magnetoelectric effects

In a multiferroic material, the ME coupling effects are expected to be enhanced. The MD effect of Dy-SMM was checked first. The dielectric permittivity out of the (0–11) plane as a function of the magnetic field applied perpendicular to the (0–11) plane at 2 K is shown in Fig. 3a. When the magnetic field was scanned from 50 kOe to −50 kOe, the dielectric permittivity showed three clear peaks. Interestingly, the magnetic field of these dielectric peaks coincide with those of RQTM. Similarly, when the magnetic field was scanned from −50 kOe to 50 kOe, three dielectric peaks also appear at the positions of RQTM (Fig. 3b). The loss tangent was measured simultaneously with the dielectric permittivity. The small value of the loss tangent (~0.002) confirmed the good insulativity of the sample and excluded the effect of extrinsic factors such as trapped space charges on the observed dielectric peaks (Supplementary Fig. 13). Moreover, clear peaks in

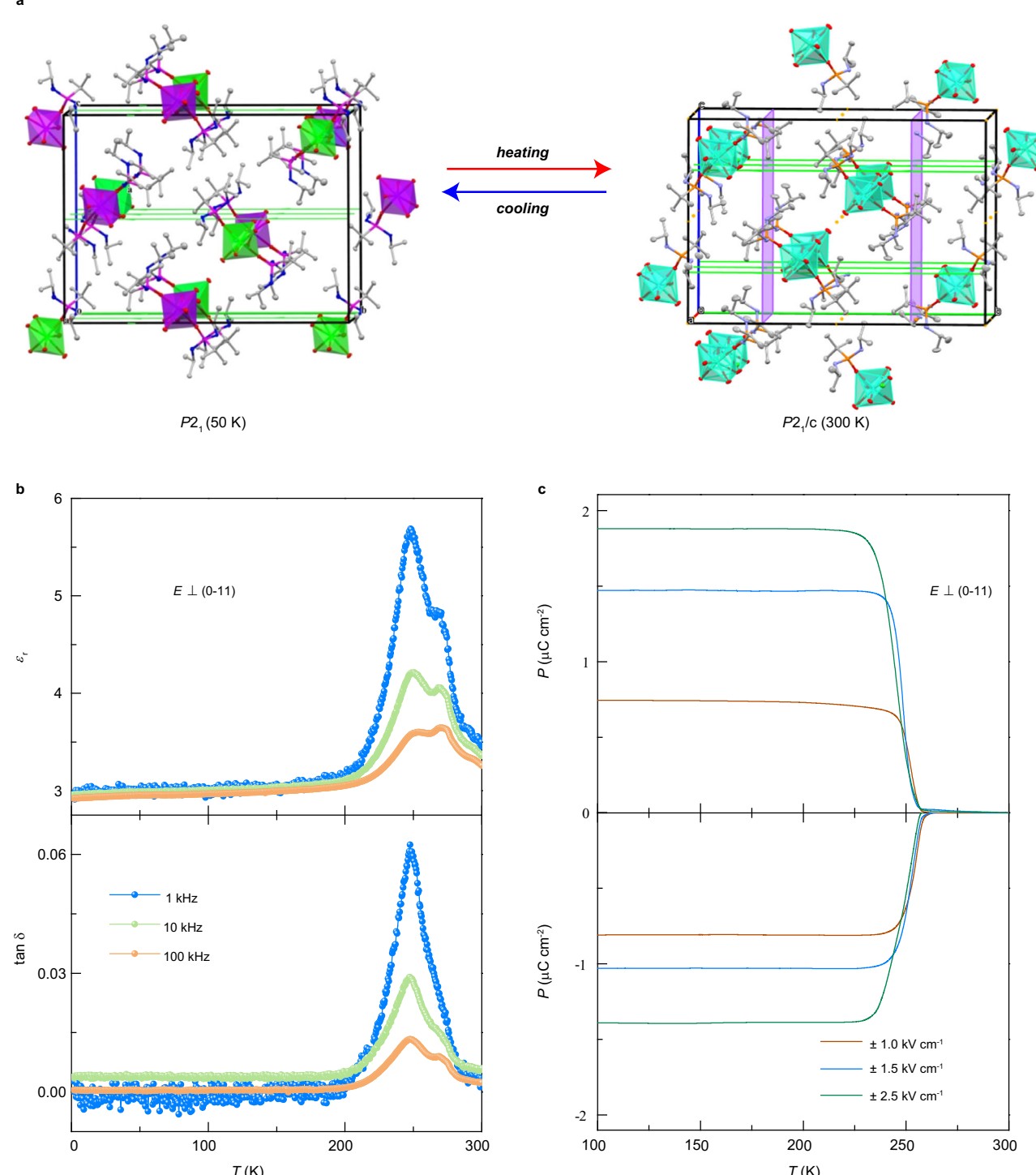

**Fig. 1 | Crystalline structure and ferroelectricity of Dy-SMM. a** Crystal structure and symmetry at 50 K (left) and 300 K (right). The green lines and lavender planes represent the symmetry elements of the $2_1$-screw axis and $c$ glide planes, respectively. Different crystalline-independent $Dy^{3+}$ ions are distinguished by differently colored polyhedrons, with cyan (Dy1 at 300 K), violet (Dy1A at 50 K) and green (Dy1B at 50 K) corresponding to three $DyO_7$ sites with different $D_{5h}$ coordination geometries. The remaining ions and atoms are omitted for clarity. **b** Dielectric permittivity and loss tangent as a function of temperature. The peak around 250 K indicates a paraelectric-ferroelectric transition. **c** Electric polarization as a function of temperature. The electric fields are applied perpendicular to the (0–11) plane.

dielectric loss are observed at the positions of the dielectric peaks, suggesting that the RQTM also affects the dielectric loss. Additionally, after an $E$-field of 4 kV/cm was applied on the sample from 300 to 2 K, the MD behavior was completely changed (Supplementary Fig. 14), suggesting that the magnetic state was modified by the applied electric field. Based on this resonant MD effect, RQTM can be detected by a

simple dielectric measurement because a dielectric peak appears whenever RQTM occurs.

Then, the mechanism of this resonant MD effect is analyzed. The tunneling of magnetization corresponds to a sudden change in angular momentum. This leads to an opposite change in phonon angular momentum due to the conservation of total angular momentum in the

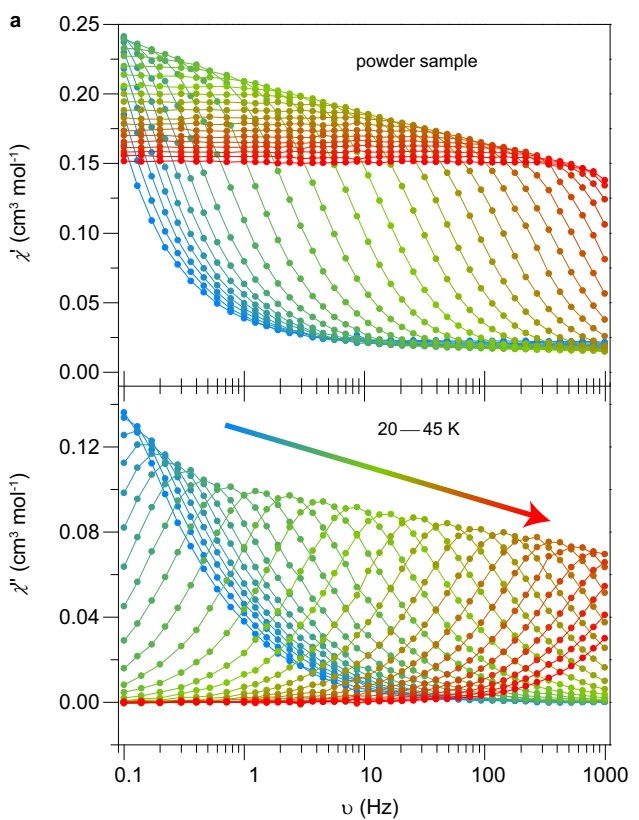

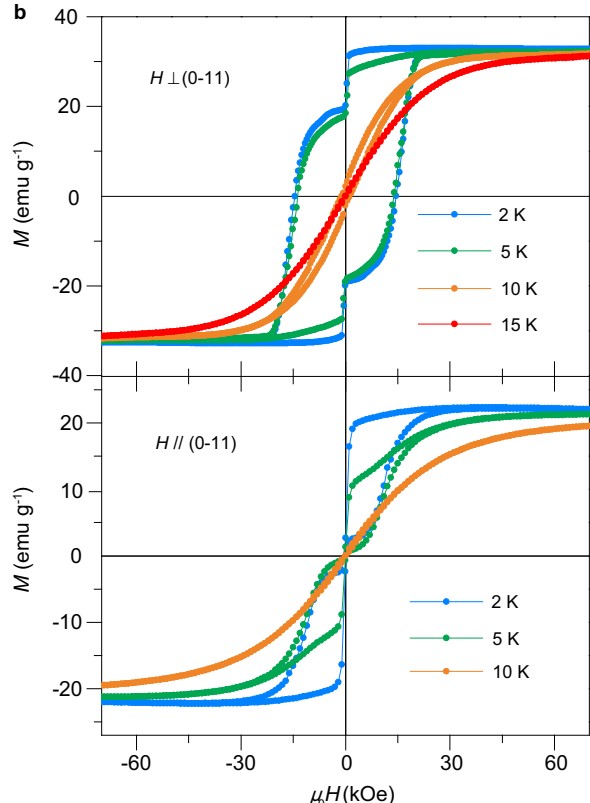

**Fig. 2 | AC magnetic susceptibility and magnetic anisotropy of Dy-SMM.**
**a** Frequency dependence of the in-phase ($\chi'$) and out-of-phase ($\chi''$) components of the AC magnetic susceptibility for the powder sample of Dy-SMM. **b** Magnetic hysteresis loops at selected temperatures for $H$ perpendicular (top) and parallel (bottom) to the (0–11) plane.

whole system, known as the Einstein–de Haas effect[30]. The abrupt change in phonon modes (lattice) is expected to cause a sizable anomaly in dielectric permittivity. Concurrently, the sudden change in magnetization with applied magnetic fields also corresponds to a reduction in the energy of the magnetic system, which is transferred to the lattice through damping. Subsequently, the sudden increase in the lattice energy may induce an abrupt change in dielectric permittivity. Both mechanisms could contribute to the observed striking MD effect.

In addition to the MD effect, the direct ME effect – magnetic field control of electric polarization – was also observed in this ferroelectric SMM. The pyroelectric peak at approximately 250 K was greatly suppressed upon application of a low magnetic field of 2 kOe (Fig. 3c, d). When a 5 kOe field was applied, the pyroelectric peak almost disappears. Correspondingly, the electric polarization rapidly decreases with increasing magnetic field, demonstrating a prominent direct ME effect. The mechanism of this low-field direct ME effect is discussed in terms of the origin of ferroelectricity. According to the structural analysis, there is no off-center displacement of $Dy^{3+}$ ions in the $DyO_7$ bipyramid units. The ferroelectricity is ascribed to the surrounding ligands which become asymmetric below 250 K. When a magnetic field is applied, the magnetoelastic effect in the $DyO_7$ bipyramid units causes a movement of the surrounding flexible ligands, and induces a change in electric polarization.

**Converse magnetoelectric effect**

The converse ME effect, i.e., electric-field control of magnetism, is more important than the direct ME effect for the potential applications of multiferroics. In the following, we demonstrate electric-field modulation of the magnetism of Dy-SMM in a series of experiments (Supplementary Fig. 15). First, the *M–H* loop at 2 K under *E*-field was investigated (Fig. 4a). The sample was cooled from 300 K to 2 K with *E*-

field off. When the temperature was stable at 2 K, the *M–H* loop was collected. Then, turning on the *E*-field, the *M–H* loop at 2 K was collected under a setting *E*-field. The open stair-shaped *M–H* loop under zero electric field is gradually reduced to a narrow loop without clear steps with increasing *E*-field; under a moderate *E* field of 3 kV/cm, the stair-shaped feature of RQTM disappears. Although the saturation magnetization changes little with increasing *E*-field, both the magnetic coercivity and remanent magnetization rapidly decrease (Fig. 4b). Moreover, the *M–H* loop along the hard direction (*H* parallel to (0–11)) at 2 K changes little with an *E*-field of 3.0 kV/cm (Supplementary Fig. 16).

After confirming the significant ME effect at 2 K, we investigated the temperature-dependence of the ME effect (Supplementary Fig. 17). Below the magnetic blocking temperature ~ 12 K, the *M–H* hysteresis loop is clearly modified – both the magnetic coercivity and saturation moments were reduced by the applied *E*-field (Supplementary Fig. 18). Above the blocking temperature, although the hysteresis loop disappears, the magnetization is still suppressed by applied *E*-field. The ME effect diminishes with increasing temperature and becomes negligible above the ferroelectric transition temperature. Meanwhile, the temperature-dependent magnetization (ZFC and FC) curves after *E*-field poling are significantly different from those without *E*-field. The magnetic blocking behavior disappeared after applying an *E*-field, which suggests the intrinsic nature of the ME effect because the change of the ZFC/FC magnetization can not be produced by a simple heating effect (Fig. 4c).

We also investigated the influence of pressure on the *M-H* loop at 2 K. As shown in Fig. 4d, the open loop becomes narrower under pressure, similar to the behavior under an electric field. This result reveals that the role of an applied electric field is equivalent to that of pressure. Thus, the microscopic mechanism of the converse ME effect

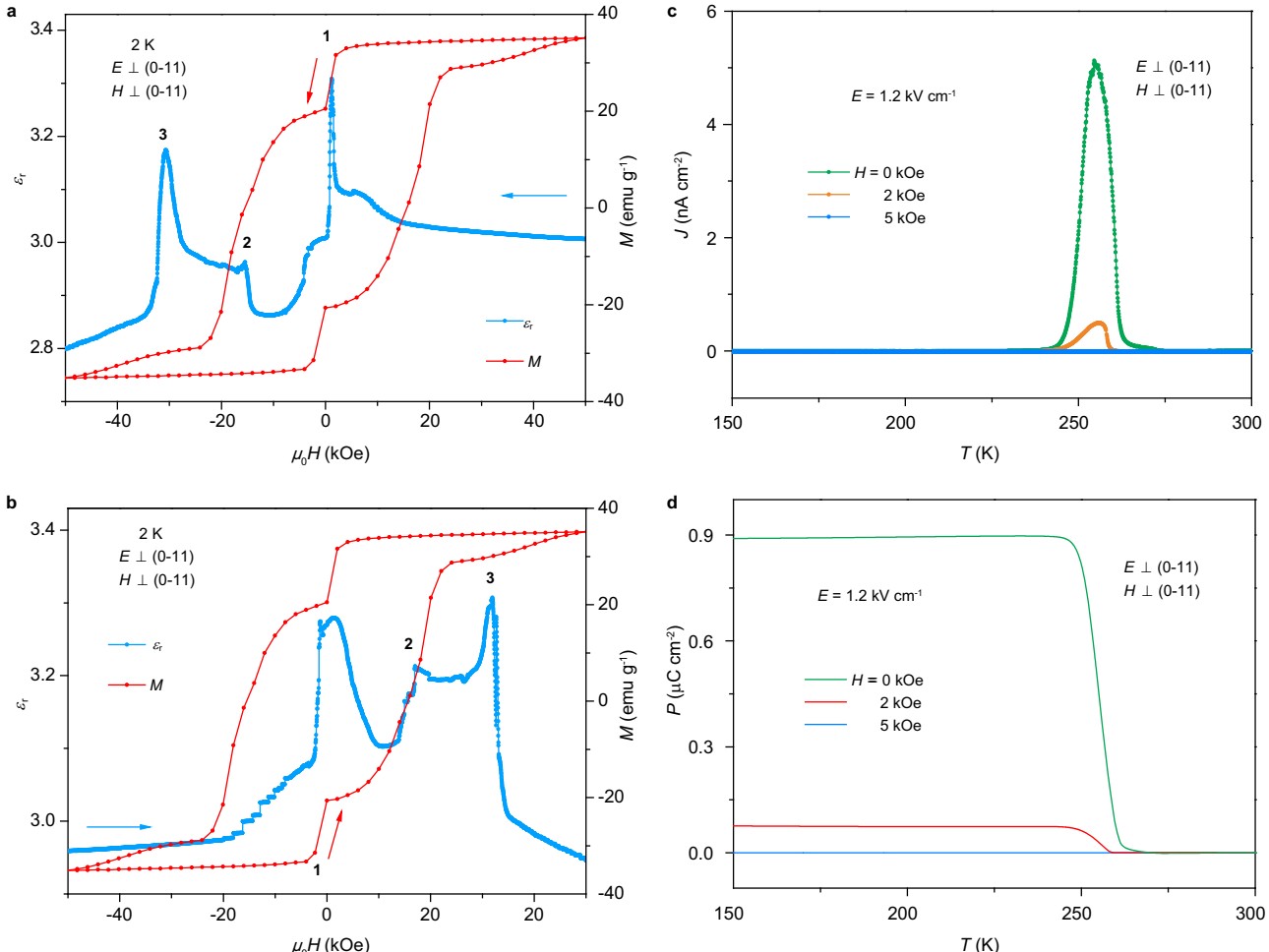

**Fig. 3 | Magnetodielectric and direct magnetoelectric effects. a** Dielectric permittivity perpendicular to the (0–11) plane with $f$ = 20 kHz as a function of decreasing magnetic field from 50 to −50 kOe at 2 K. The arrows indicate the direction of the sweeping magnetic field. The $M$–$H$ loop at 2 K is also plotted for comparison. **b** Dielectric permittivity as a function of increasing magnetic field from −50 to 50 kOe at 2 K. The arrows indicate the direction of the sweeping

magnetic field. The $M$–$H$ loop at 2 K is also plotted for comparison. Three dielectric peaks appear at the RQTM positions due to the ME coupling. **c** Pyroelectric current as a function of temperature under several magnetic fields. **d** Electric polarization as a function of temperature under several magnetic fields. The electric polarization rapidly decreases with increasing magnetic field.

is likely associated with the piezoelectric effect of the ferroelectric SMM.

## Electric field control of magnetic dynamics

The AC magnetic susceptibility in the low temperature and low frequency ranges (2–30 K, <1 Hz) were remarkably altered with $E$-field. In particular, the sharp peaks in $\chi'$ and $\chi''$ were broadened after $E$-field poling (Supplementary Fig. 19). Thus, not only the static magnetization but also the magnetic dynamics of this SMM can be modulated by electric fields. However, in the high temperature range, the frequency-dependent AC magnetic susceptibility after $E$-field poling was evaluated and yielded a slightly reduced energy barrier $U_{eff}$ = 613(6) K (Supplementary Figs. 20 and 21, and Supplementary Table 7). Therefore, these results suggest that applied $E$-fields have a strong influence on the RQTM at low temperatures but little influence on the thermally excited Orbach and Raman relaxation processes at high temperatures. This may be due to the less influence of the applied $E$-fields on the large energy barrier of the second excited state.

Next, we investigated more closely the influence of the $E$-field on the magnetic relaxation at 2 K. First, a high magnetic field was applied to saturate the magnetization, and then the electric field is turned on and the magnetic field was cut off, after that the remanent

magnetization ($M_r$) was measured as a function of time under $E$-field. The magnetization relaxation was fitted with the exponential function $M_r = a*exp^{(-t/\tau)} + b_0$, which is anticipated for a collection of identical particles with a single energy barrier. The relaxation time ($\tau$) determined by the function fit strongly depends on the applied $E$-field. It decreased from 16983 s in zero electric field to 262 s under an $E$-field of 3 kV/cm (Fig. 5b), which indicates that the magnetic relaxation was dramatically accelerated under an $E$-field. This enhancement of magnetic relaxation indicates that the magnetic anisotropy energy was suppressed by applied $E$-fields, consistent with the decrease in the coercive field under $E$-field. The detailed fitting results are presented in Supplementary Figs. 22 and 23.

To compare the role of applied $E$-fields with that of temperature, we measured the magnetic relaxation without $E$-field in the temperature range of 2 K to 10 K. The relaxation time decreases from 16983 s at 2 K to 72 s at 10 K. Remarkably, the relaxation time of $\tau$ = 262 s at 2 K under $E$ = 3.0 kV/cm is approximately equal to the result of $\tau$ = 223 s above 8 K without an applied $E$-field. It means that the influence of an $E$-field of 3.0 kV/cm is equivalent to the temperature increasing from 2 K to 8 K. Therefore, the change of magnetic relaxation time with $E$-fields is due to the intrinsic ME effect of the Dy-SMM, rather than being a result of a heating effect. The excluding of heating effect is discussed in detail in supplementary discussion (Supplementary Figs. 24 and 25).

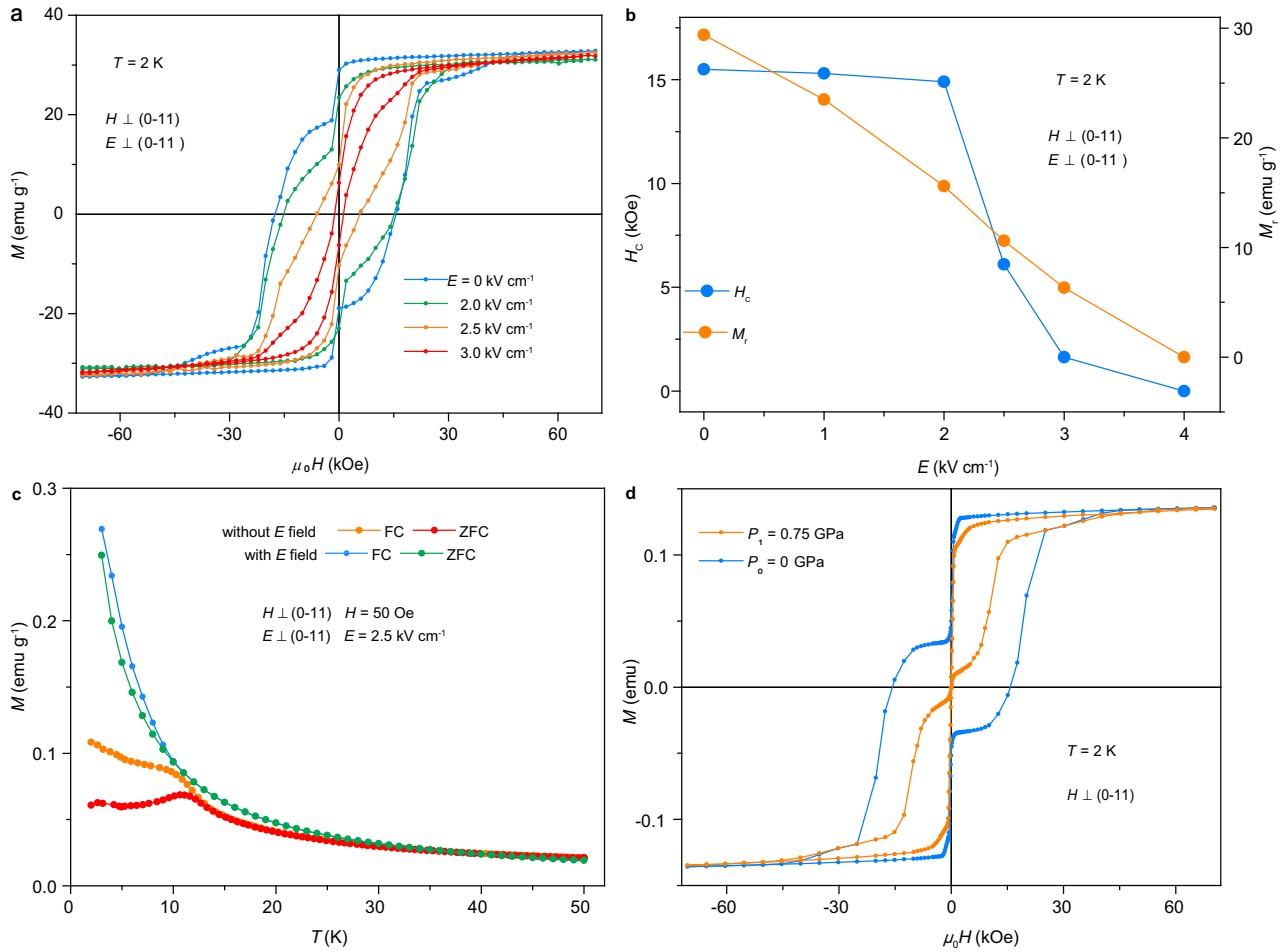

**Fig. 4 | Electric field control of magnetization. a** The $M–H$ loops at 2 K measured under different electric fields. Both the electric field and magnetic field are applied perpendicular to the (0–11) plane. **b** Magnetic coercivity ($H_C$) and remanent magnetization ($M_r$) as a function of the applied electric field. Both $H_C$ and $M_r$ decrease with increasing $E$ field. **c** Comparison of the $M–T$ curves measured with and without an electric field. The magnetic blocking behavior disappears after $E$–field poling. **d** Influence of pressure on the $M–H$ loop at 2 K. The magnetic field is applied perpendicular to the (0–11) plane. The role of the electric field is similar to that of pressure.

## Discussion

The mechanism of $E$-field control of magnetism in this Dy-SMM is believed to be closely related to the piezoelectric effect in the ferroelectric phase. An applied electric field would generate a deformation of the lattice and induce changes in the crystal field surrounding $Dy^{3+}$ ions. Subsequently, the magnetic anisotropy of $Dy^{3+}$ would be modified. The observed decrease of magnetic hysteresis and disappearance of magnetic blocking under external $E$-fields indicate that the energy barrier of magnetic anisotropy along [0–11] is lowered upon application of an electric field. The magnetic anisotropy of Dy-SMM is related to the structure of the $DyO_7$ bipyramidal units as well as the dipole–dipole interaction between $Dy^{3+}$ ions. The lattice change induced by the piezoelectric effect could be due to either distortion of the $DyO_7$ pentagonal bipyramids or the distance between the $Dy^{3+}$ ions. However, the former requires much more energy than the latter. Under a low $E$-field, the alignment of FE domains is more likely to be accompanied by the rotation and stretching of the organic ligands surrounding the $DyO_7$ bipyramids, leading to a change in the distance between the $DyO_7$ bipyramids. As a result, the dipole–dipole interaction between $Dy^{3+}$ ions is weakened and the magnetic anisotropy of $Dy^{3+}$ ions is modified by an applied electric field. Under a high $E$-field, the distortion of the $DyO_7$ bipyramids may become notable, which also contributes to the alteration of the magnetic anisotropy. The observed $E$-field control of macroscopic magnetization and magnetic dynamics

is a consequence of modulation of the single-ion magnetic anisotropy of $Dy^{3+}$ by applied $E$-fields.

To observe the influence of $E$-field on the crystal lattice, the lattice parameters were quantitatively characterized by X-ray diffraction with an applied $E$-field perpendicular to the (0–11) plane. The lattice parameters $b$ (from 25.52 to 25.66 Å) and $c$ (from 18.44 to 18.51 Å) change more strongly than $a$-axis (from 9.00 to 9.01 Å) when an $E$-field of 2 kV/cm is applied (Fig. 6a) The peak of a randomly selected diffraction pattern revealed a remarkable change upon the application of $E$-field, with a clear shift within a unit cell (Fig. 6b, c). In addition to the shift in position, the intensity of the Bragg peak is apparently reduced by applied $E$-fields (Fig. 6d). Other Bragg peaks exhibit similar changes in the pattern and intensity with applied $E$-fields, which reflects the shifts in atom positions[31]. The above results confirmed that the lattice is modified by electric field. Subsequently, the change of magnetism under $E$-fields is due to the lattice change through the piezoelectric effect.

The achievement of significant ME effects in Dy-SMM evidences a promising field of ferroelectric SMMs. By introducing ferroelectricity into SMMs, the quantum behaviors of magnetism become electrically tunable and detectable through the ME coupling, which is critical for the applications of SMMs. Especially, spin qubit based on ferroelectric SMMs could be a potential candidate for solid-state quantum computing. The ME coupling in the ferroelectric SMMs would enable

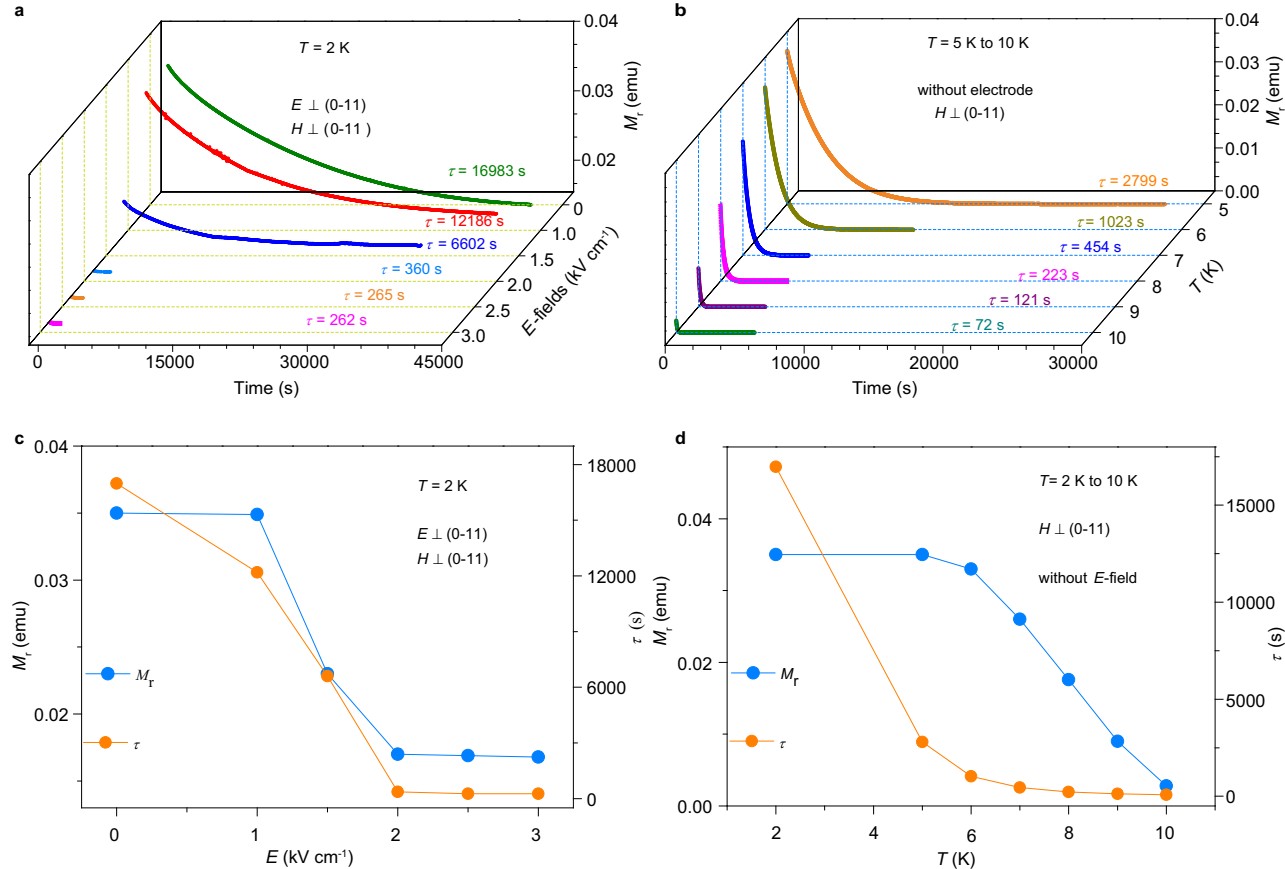

**Fig. 5 | Electric-field modulation of magnetic relaxation.** Both the electric field and magnetic field were applied perpendicular to the (0–11) plane. **a** Magnetization relaxation under different $E$-fields (0, 1.0, 1.5, 2.0, 2.5 and 3.0 kV/cm) at 2 K. The magnetic relaxation time ($\tau$) was obtained from the best fit with the exponential decay function. **b** Magnetization decay vs. time in the temperature range of 5–10 K without an $E$-field. **c** The parameter $M_r$ and $\tau$ as a function of applied $E$-field at 2 K. **d** Temperature dependence of $M_r$ and $\tau$ without an $E$-field. The $\tau = 262$ s at 2 K under 3.0 kV/cm is close to the $\tau = 223$ s at 8 K without an $E$-field.

writing the qubit with an electric field and read the qubit by a simple dielectric measurement. In the future, more efforts are needed to further improve the ferroelectricity and ME coupling in SMMs through a careful design of the structures and selection of magnetic ions as well as the organic ligands.

## Methods

### Synthesis of the materials

The starting materials were purchased from Sigma–Aldrich and were used without further purification unless otherwise noted. DyCl$_3$·6H$_2$O (9.4 mg, 0.25 mmol) was slowly added to a solution of the phosphoric diamide ligand $^t$BuPO(NH$^i$Pr)$_2$ (12.2 mg, 0.55 mmol) in acetonitrile (50 mL). The mixture was refluxed for 2.5 h at 85 °C and then transferred to a vial after filtration. The filtrate was stirred and then kept still at 40 °C until the volume was reduced to nearly 25 mL due to evaporation. Slightly irregular rectangular colorless crystals were obtained from this crystallization solution after three days at room temperature (18 °C). The single-crystal sizes ranged from approximately 0.5 mm × 0.2 mm × 1.0 mm to 6.0 mm × 4.0 mm × 2.0 mm, depending on the evaporation temperatures and air humidity levels.

### Structural characterization

Single-crystal X-ray diffraction data at different temperatures were collected on a Bruker APEX-III area-detector diffractometer (50 K and 200 K) and an Oxford SuperNova diffractometer (300 K). The crystal structures were determined based on direct methods and refined by the full-matrix least-squares method on F$^2$ with anisotropic thermal parameters for all nonhydrogen atoms using the SHELXS-97 and

SHELXS-97 programs. Hydrogen atoms were located geometrically and refined isotropically. These single-crystal X-ray data have been deposited in the Cambridge Crystallographic Data Centre (www.ccdc.cam.ac.uk/data_request/cif): CCDC-1984441 (50 K), 1984442 (200 K) and CCDC-1984443 (300 K). The lattice parameter change under electric fields was measured on a Bruker APEX-III area-detector diffractometer at 50 K. A Keithley 6517B electrometer was used to supply the DC voltage on the single crystal sample with two electrodes made by painting silver paste on the (0–11) planes. Thermogravimetric analysis (TGA) was performed on a Netzsch TG 209 TG-DTA analyzer from 40 °C to 800 °C under a nitrogen atmosphere with a heating rate of 10 °C/min.

Cryogenic luminescence experiments were conducted in a cryostat (Montana model C2) at temperatures ranging from 5 K to 295 K. A Princeton SP2500 spectrometer equipped with a nitrogen-cooled silicon charge-coupled device camera was used to record the luminescence. Before the signal light entered the spectrometer, the fundamental frequency light was filtered by a 532 nm longpass filter. The integration time of the spectrometer was 5 s.

### Magnetic properties measurements

The magnetic properties on both single-crystal and powder samples were measured using a Quantum Design superconducting quantum interference device magnetometer (MPMS-XL7). Electric field control of magnetism was measured on single-crystal samples with a typical size of 3.0 mm × 1.0 mm × 0.5 mm using a sample probe shown in Supplementary Fig. S15. The electrodes were made by painting silver paste on the (0–11) planes and a Keithley 6517B electrometer was used

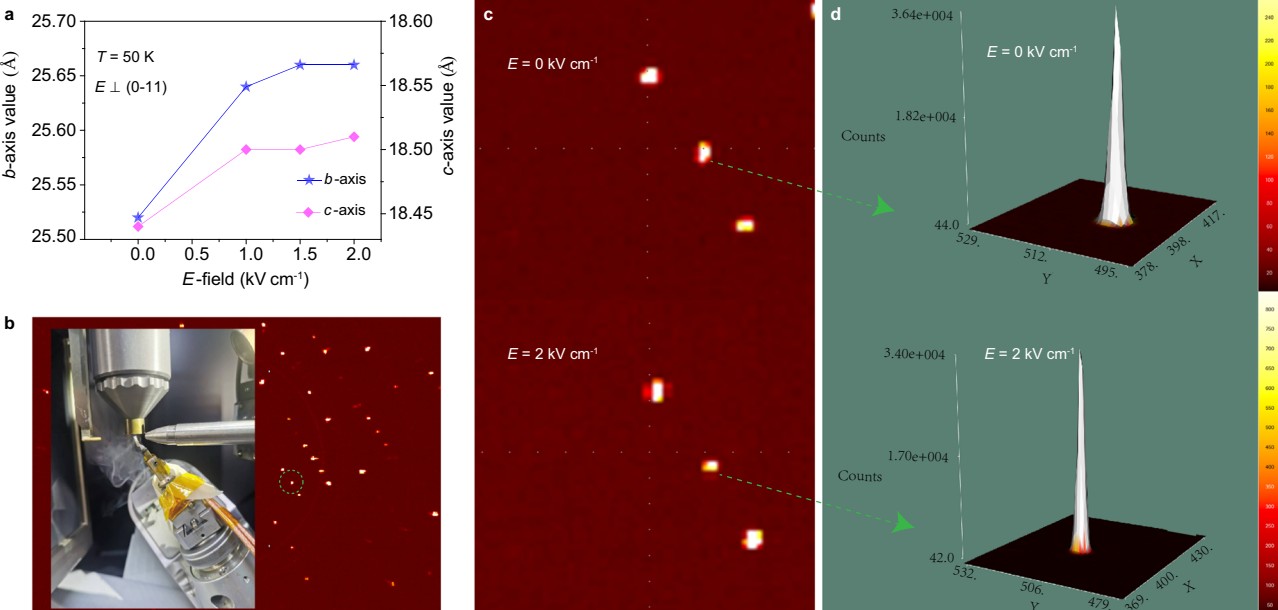

**Fig. 6 | Single-crystal X-ray diffraction of Dy-SMM under electric fields applied perpendicular to the (0–11) plane at 50 K. a** The variation of the lattice parameters under different *E*-fields (0, 1.0, 1.5 and 2 kV/cm). **b** Representative X-ray diffraction pattern of a single crystal. The circled Bragg peak is (1,5,−3). Inset: a sample with wires and electrodes, and cryogenic gas feed. **c** Comparison of the location of (1,5,−3) peak with and without *E*-fields. **d** Comparison of the intensity (counts) of (1,5,−3) peak with and without *E*-fields.

as to apply the DC voltage on the sample. In addition, a control experiment of *E*-field modulation of the *M–H* loop was performed by placing the sample between two parallel conductive indium tin oxide (ITO) films without direct contact with the sample.

### Dielectric and pyroelectric measurements

All electrical measurements were performed on a Quantum Design Physical Property Measurement System (PPMS). The samples for dielectric and pyroelectric measurements were polished with sandpaper into thin plates with a typical size of 3.0 mm × 1.0 mm × 0.5 mm, and electrode contacts were made from silver paste painted on the (0–11) planes. The dielectric permittivity was measured with an Andeen Hagerling 2700 capacitance bridge. The frequency and external AC voltage intensity were 20 kHz and 1 V, respectively. The pyroelectric current was collected by a Keithley 6517B electrometer with increasing temperature at a warming rate of 2 K/min. The electric polarization was obtained by integrating the pyroelectric current over time.

### Ab initio calculations

The ab initio calculations were of the CASSCF/RASSI/SINGLE_ANISO type. They were performed on the structures obtained with the XRD analysis using the program OPENMOLCAS 21.02. The basis sets were chosen from the ANO-RCC library. Dy atoms were treated with the VQZP basis set; O atoms were treated with the VTZP basis set; N, P, C and Cl atoms were treated with the VDZP basis set; and H atoms were treated with the VDZ basis set. The state-averaged CASSCF orbitals of the sextets, quartets and doublets were optimized with 21, 224 and 490 states, respectively. Furthermore, all spin sextet states, 128 spin quartet states and 130 spin doublet states were chosen to construct and diagonalize the spin-orbit (SO) coupling Hamiltonian with the RASSI module. These computed spin-orbit states were further used by the SINGLE_ANISO program for computation of the g-tensors, crystal field parameters and magnetic energy levels for the doublets of the ground $J = 15/2$ multiplet of the $^6H_{15/2}$ term for Dy(III).

## Data availability

Data generated or analyzed during this study are included in the Article and its Supplementary Information files. The crystallographic data for the structures reported in this Article have been deposited at the Cambridge Crystallographic Data Centre, under deposition numbers CCDC 1984441 (for Dy-SMM at 50 K), 1984442 (for Dy-SMM at 200 K) and 1984443 (for Dy-SMM at 300 K). Copies of the data can be obtained free of charge via https://www.ccdc.cam.ac.uk/structures/.

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

## Acknowledgements

Financial support for this project was partially provided by the National Key R&D Program of China (grant ref: 2021YFA1400300), the National Natural Science Foundation of China (grant refs: 21931004, 12227806 and 21973049), the Ministry of Education of China (grant refs: B12015 and 2023CDJXY-049), the China Postdoctoral Science Foundation funded project (grant ref: 2019M651009) and Frontiers Science Center for New Organic Matter, Nankai University (grant ref: 63181206). We genuinely appreciate for the discussions with Prof. Sang-Wook Cheong (Center for Quantum Materials Synthesis, Rutgers University).

## Author contributions

The concept for the project was initially developed by Y.X.W., Y.S and P.C.. Y.X.W. synthesized the material. Y.X.W., D.S and Y.M. performed the magnetic and electrical measurements. Y.S and P.C. interpreted the experimental data while Y.X.W., Y.S. and P.C. wrote the paper. All authors discussed and provided input on the manuscript.

## Competing interests

The authors declare no competing interests.
