## [Peer Review File · Nature Communications]

REVIEWER COMMENTS

Reviewer #1 (Remarks to the Author):

Sun, Cheng and co-workers characterize a dysprosium single-molecule magnet (SMM) both using the standard magnetic characterization methods and using a wide variety of electric characterization methods. While from a magnetic point of view the SMM is rather average, the characterization of the various electric properties goes well beyond the usual and demonstrates ground-breaking science that is, in my opinion, suitable for publication in Nature Communications. I have strong reservations about the main thesis of the manuscript, namely the concept of the system as multiferroic SMM (see below); however, even if this main thesis abandoned, the manuscript still provides enough ground-breaking results to validate its publication. In any case, while presenting fascinating results, the manuscript requires significant revision.

The authors describe the characterized SMM material as a multiferroic SMM. The definition of such a system is, however, somewhat ambiguous. A conventional multiferroic material demonstrates ferroelectricity and ferromagnetism in the same phase. The onset of SMM behavior, however, is not related to any specific phase change but rather to the timescale of the relaxation dynamics within a single paramagnetic phase. Thus, SMM behavior is not ferroic behavior in the same sense as ferromagnetism is, as the latter is related to a specific phase whereas the former is related to the timescale of relaxation. I think that the most loose definition of a multiferroic SMM material should at least include the observation of slow magnetization dynamics and ferroelectricity in the same macrostate as defined by the temperature and other state variables.

The authors demonstrate a transition to a ferroelectric phase below at about 250 K, and based on Figure 5, there is no other transition between 100 K and 250 K. The magnetic blocking temperature was determined as 12 K based on the divergence of ZFC and FC magnetization. Based on Figure 2, slow relaxation dynamics were observed using ac susceptibility up to 45 K. It is not clear from the measurements whether the ferroelectric phase is retained down to the temperatures where slow relaxation is observed. Thus, based on the reported data, it is not possible to state that the system is a multiferroic SMM even within the most loose definition.

I see two possible approaches for the authors to bring the manuscript to a state that is publishable in Nature Communications:

- 1) i) Demonstrate experimentally that both the slow relaxation and ferroelectricity coexist under the same set of state variables (temperature, magnetic field etc.). Preferably, the ferroelectricity should be demonstrated below the magnetic blocking temperature of the SMM (as there is no universally accepted definition, there is some leeway in this temperature). ii) More thoroughly discuss what exactly

can be considered as a multiferroic SMM, what is the authors' exact definition (both physical and mathematical) for such a system, and how this definition is related and how it differs from the definition of conventional multiferroic materials.

2) Disregard the multiferroic aspect of the manuscript and focus on the other fascinating results relating to the interplay of electric and magnetic properties in the system.

In addition to the above considerations, I also found the following explanation for the correlation of the dielectric peaks with the QTM a bit puzzling:

“In terms of the conservation of total angular momentum in the whole system, the surrounding lattice experiences a spin-transfer torque that produces an opposite change in phonon angular momentum, known as the Einstein–de Haas effect. In a multiferroic phase, the tunnelling of magnetization can induce a sizable change in the dielectric permittivity due to the strong spin-phonon coupling. Based on this resonant magnetodielectric effect, RQTM can be detected by a simple dielectric measurement because a dielectric peak appears whenever RQTM occurs.”

This explanation should be elaborated considerably as now it seems more like a hypothesis than an explanation. This correlation of QTM and dielectric peaks is a very interesting observation and the manuscript would be greatly enhanced if it was discussed in a more detailed manner, and even better if the authors could provide a mathematical model to explain the observation. I should also note that the Einstein–de Haas effect is not widely known in the SMM community and a more thorough explanation of it would be useful.

Reviewer #2 (Remarks to the Author):

The authors report the successful creation of a class of molecular materials, multiferroic SMMs, by introducing ferroelectricity into a dysprosium-based SMM through spatial cooperation between flexible organic ligands and halide ions. Ferroelectricity as well as strong ME effects are demonstrated in millimetre-sized single crystals. A low magnetic field is able to suppress electric polarization, and the magnetic properties are effectively modulated by applying electric fields. It is a new class of molecular materials but the performance seems not to be outstanding for practical applications, so at present I am a little reluctant to recommend its publication, unless the authors can reveal any interesting physics.

1. Not all polar structures are ferroelectric, and the origin of ferroelectricity is yet to be clarified, especially how the polarization can be switched: movement of Dy ions? proton hopping? rotation of ligands? van der Waals sliding?

2. The origin of the coupling between ferroelectricity and magnetism is also yet to be revealed.

3. Why the authors call them single-molecule magnets? In my view, they are weakly-bonded crystals of molecular magnets, while each so-called single-molecule magnet should be able to store 1 bit data independently.

Reviewer #3 (Remarks to the Author):

The manuscript "Electrical detection and direct modulation of magnetism in a Dy-based multiferroic SMM" is a very nice new material and describes the synthesis, crystal structure determination and physical properties (magnetic, magnetoelectric) investigation of Dy(III) complex. The authors demonstrated the influence of an applied electric field on magnetic properties and the influence of an applied magnetic field on dielectric/pyroelectric ones. The demonstration of such a mutual influence of magnetic and electric properties in molecular materials is new. The manuscript is well written and the physical properties are well investigated. The results are of interest for a broad community and of high relevance to be published in high rank journals, as Nature Communications. In order to make the text more concise of magnetoelectric couplings, I suggest the author to move the heating effect discussion to supplementary.

Response to reviewers' comments

Reviewer #1 (Remarks to the Author):

Sun, Cheng and co-workers characterize a dysprosium single-molecule magnet (SMM) both using the standard magnetic characterization methods and using a wide variety of electric characterization methods. While from a magnetic point of view the SMM is rather average, the characterization of the various electric properties goes well beyond the usual and demonstrates ground-breaking science that is, in my opinion, suitable for publication in Nature Communications. I have strong reservations about the main thesis of the manuscript, namely the concept of the system as multiferroic SMM (see below); however, even if this main thesis abandoned, the manuscript still provides enough ground-breaking results to validate its publication. In any case, while presenting fascinating results, the manuscript requires significant revision.

The authors describe the characterized SMM material as a multiferroic SMM. The definition of such a system is, however, somewhat ambiguous. A conventional multiferroic material demonstrates ferroelectricity and ferromagnetism in the same phase. The onset of SMM behavior, however, is not related to any specific phase change but rather to the timescale of the relaxation dynamics within a single paramagnetic phase. Thus, SMM behavior is not ferroic behavior in the same sense as ferromagnetism is, as the latter is related to a specific phase whereas the former is related to the timescale of relaxation. I think that the most loose definition of a multiferroic SMM material should at least include the observation of slow magnetization dynamics and ferroelectricity in the same macrostate as defined by the temperature and other state variables.

The authors demonstrate a transition to a ferroelectric phase below at about 250 K, and based on Figure 5, there is no other transition between 100 K and 250 K. The magnetic blocking temperature was determined as 12 K based on the divergence of ZFC and FC magnetization. Based on Figure 2, slow relaxation dynamics were observed using ac susceptibility up to 45 K. It is not clear from the measurements whether the ferroelectric phase is retained down to the temperatures where slow relaxation is observed. Thus, based on the reported data, it is not possible to state that the system is a multiferroic SMM even within the most loose definition.

I see two possible approaches for the authors to bring the manuscript to a state that is publishable in Nature Communications:

1) i) Demonstrate experimentally that both the slow relaxation and ferroelectricity coexist under the same set of state variables (temperature, magnetic field etc.). Preferably, the ferroelectricity should be demonstrated below the magnetic blocking temperature of the SMM (as there is no universally accepted definition, there is some leeway in this temperature). ii) More thoroughly discuss what exactly can be considered as a multiferroic SMM, what is the authors' exact definition (both physical and mathematical) for such a system, and how this definition is related and how it

differs from the definition of conventional multiferroic materials. 2) Disregard the multiferroic aspect of the manuscript and focus on the other fascinating results relating to the interplay of electric and magnetic properties in the system.

Response:

We sincerely appreciate the reviewer's valuable comments and instructive suggestions regarding the definition of "multiferroic SMM." We agree with the reviewer that the term "multiferroic SMM" is somewhat misleading because the magnetism of SMMs does not belong to any magnetic ordering (ferroic) state. Thus, we decide to change the term "multiferroic SMM" to "ferroelectric SMM".

In our study, we have measured the dielectric permittivity and pyroelectric current down to 2 K (though we merely showed the data between 100 and 300 K in Fig. 1), and observed only one dielectric anomaly and a pyroelectricity peak around 250 K within the temperature range of 2-300 K. Moreover, the variable-temperature luminescence spectra from 5 to 295 K also indicates that there is no other structural phase transition at low temperatures (see Supplementary Information Fig. 4). Thus, the ferroelectricity is confirmed to persist below the magnetic blocking temperature of the SMM. We have modified Fig. 1b and 1c to show the full data between 2 and 300 K in the revised paper.

In addition to the above considerations, I also found the following explanation for the correlation of the dielectric peaks with the QTM a bit puzzling:

"In terms of the conservation of total angular momentum in the whole system, the surrounding lattice experiences a spin-transfer torque that produces an opposite change in phonon angular momentum, known as the Einstein–de Haas effect. In a multiferroic phase, the tunnelling of magnetization can induce a sizable change in the dielectric permittivity due to the strong spin-phonon coupling. Based on this resonant magnetodielectric effect, RQTM can be detected by a simple dielectric measurement because a dielectric peak appears whenever RQTM occurs."

This explanation should be elaborated considerably as now it seems more like a hypothesis than an explanation. This correlation of QTM and dielectric peaks is a very interesting observation and the manuscript would be greatly enhanced if it was discussed in a more detailed manner, and even better if the authors could provide a mathematical model to explain the observation. I should also note that the Einstein–de Haas effect is not widely known in the SMM community and a more thorough explanation of it would be useful.

Response:

Following the reviewer's suggestion, we have improved the explanation on the correlation of the dielectric peaks with the QTM. The Einstein–de Haas effect demonstrates the rotation of a suspended magnet when its magnetization is changed. It is a consequence of the conservation of angular momentum. The change in magnetization corresponds to a change in angular momentum, and it is compensated by an opposite change in the phonon angular momentum in order to make the conservation of the total angular momentum. The abrupt change in phonon modes (lattice) is expected to cause a sizable anomaly in dielectric permittivity. Meanwhile,

the sudden change in magnetization with applied magnetic fields also corresponds to a reduction in the energy of the magnetic system, which is transferred to the lattice through damping. Subsequently, the sudden increase in the lattice energy may induce an abrupt change in dielectric permittivity. Both mechanisms could contribute to the observed striking magnetodielectric effect. We have included more details in the discussion in the revised paper.

Reviewer #2 (Remarks to the Author):

The authors report the successful creation of a class of molecular materials, multiferroic SMMs, by introducing ferroelectricity into a dysprosium-based SMM through spatial cooperation between flexible organic ligands and halide ions. Ferroelectricity as well as strong ME effects are demonstrated in millimetre-sized single crystals. A low magnetic field is able to suppress electric polarization, and the magnetic properties are effectively modulated by applying electric fields. It is a new class of molecular materials but the performance seems not to be outstanding for practical applications, so at present I am a little reluctant to recommend its publication, unless the authors can reveal any interesting physics.

1. Not all polar structures are ferroelectric, and the origin of ferroelectricity is yet to be clarified, especially how the polarization can be switched: movement of Dy ions? proton hopping? rotation of ligands? van der Waals sliding?

Response:

Firstly, we want to clarify that the most promising application for SMMs is quantum computing because each SMM can act as a spin qubit. The achievement of electric-field control and electrical detection of quantum tunneling of magnetization in our study is critical for the practical applications of SMMs at low temperatures.

Secondly, we thank the reviewer for the good question on the origin of ferroelectricity in the SMM. According to the structural analysis based on XRD data, there is no off-center displacement of Dy³⁺ ions in the DyO₇ bipyramid units. The ferroelectricity is ascribed to the surrounding ligands which become asymmetric below 250 K. The switch of electric polarization is through the movement of the ligands.

2. The origin of the coupling between ferroelectricity and magnetism is also yet to be revealed.

Response:

Although the ferroelectricity and magnetism arise independently in this SMM, the piezoelectric effect in the ferroelectric phase is able to yield a considerable ME effect. As we described in the discussion section in the manuscript, the piezoelectric effect induces a change in the distance between the DyO₇ bipyramids (low fields) and distortion of the DyO₇ bipyramids (high fields), which modifies the magnetic anisotropy energy and alter the magnetic relaxation process. In turn, the magnetoelastic effect could cause a lattice change, especially the movement of the flexible ligands, and induce a change in electric polarization. We have included more content to discuss the mechanism in the revised paper.

3. Why the authors call them single-molecule magnets? In my view, they are weakly-bonded crystals of molecular magnets, while each so-called single-molecule magnet should be able to store 1 bit data independently.

Response:

The term of “single-molecule magnets” was conceived by chemists in the 1990s after the pioneering discovery in 1993 by Prof. Sessoli and Prof. Gatteschi. Indeed, each single-molecule magnet is able to act as a classical or quantum bit, and there is only weak dipole-dipole interaction between molecules.

Reviewer #3 (Remarks to the Author):

The manuscript “*Electrical detection and direct modulation of magnetism in a Dy-based multiferroic SMM*” is a very nice new material and describes the synthesis, crystal structure determination and physical properties (magnetic, magnetoelectric) investigation of Dy(III) complex. The authors demonstrated the influence of an applied electric field on magnetic properties and the influence of an applied magnetic field on dielectric/pyroelectric ones. The demonstration of such a mutual influence of magnetic and electric properties in molecular materials is new. The manuscript is well written and the physical properties are well investigated. The results are of interest for a broad community and of high relevance to be published in high rank journals, as Nature Communications. In order to make the text more concise of magnetoelectric couplings, I suggest the author to move the heating effect discussion to supplementary.

Response:

Thank you for your evaluation and thoughtful suggestions. We have relocated the portion of the discussion on heating effect to the Supplementary Information.

Summary of changes

1. The title of the paper is modified by replacing “multiferroic” with “ferroelectric”.
2. On page 3, the introduction is slightly modified.
3. On page 5, Fig. 1b is replaced with the full data between 2 and 300 K.
4. On page 6, more description is included to clarify that ferroelectricity persists down to the lowest temperature.
5. On page 10, more discussion is included to explain the mechanism of the observed magnetodielectric effect.
6. On page 12, more discussion is included to explain the origin of ferroelectricity and mechanism of the direct ME effect.
7. On page 19, the conclusion is modified.

REVIEWERS' COMMENTS

Reviewer #1 (Remarks to the Author):

The authors have adequately addressed the issues raised in my review report, and, in my opinion, the manuscript is suitable for publication in Nature Communications.

Reviewer #2 (Remarks to the Author):

This version might be nearly accepted.

The authors may add some words about what kind of properties may enable high-performance as a multiferroic Qbit.

Reviewer #3 (Remarks to the Author):

The authors have addressed the comments I originally raised, and I believe the manuscript is highly deserving of being published in Nature Communications.

Response to reviewers' comments

Reviewer #1 (Remarks to the Author):

The authors have adequately addressed the issues raised in my review report, and, in my opinion, the manuscript is suitable for publication in Nature Communications.

Response:

Thank you for your response and recognition.

Reviewer #2 (Remarks to the Author):

This version might be nearly accepted.

The authors may add some words about what kind of properties may enable high-performance as a multiferroic Qbit.

Response:

Following the reviewer's suggestion, we have included a few words on the potential of ferroelectric single-molecule magnets as spin qubit on page 14. "Especially, spin qubit based on ferroelectric SMMs could be a potential candidate for solid-state quantum computing. The ME coupling in the ferroelectric SMMs would enable writing the qubit with an electric field and read the qubit by a simple dielectric measurement".

Reviewer #3 (Remarks to the Author):

The authors have addressed the comments I originally raised, and I believe the manuscript is highly deserving of being published in Nature Communications.

Response:

Thank you for your evaluation and support.